# Prediction of Cutting Force and Chip Formation from the True Stress–Strain Relation Using an Explicit FEM for Polymer Machining

**DOI:** 10.3390/polym14010189

**Published:** 2022-01-04

**Authors:** Bin Yang, Hongjian Wang, Kunkun Fu, Chonglei Wang

**Affiliations:** 1School of Aerospace Engineering and Applied Mechanics, Tongji University, Shanghai 200092, China; yangbin2018@tongji.edu.cn (B.Y.); 1984fukunkun@tongji.edu.cn (K.F.); 2School of Aerospace, Mechanical and Mechatronic Engineering, The University of Sydney, Sydney 2006, Australia; hong.j.wang@sydney.edu.au; 3College of Shipbuilding Engineering, Harbin Engineering University, Harbin 150001, China

**Keywords:** cutting force, chip formation, orthogonal cutting, explicit FEM

## Abstract

In the present work, an explicit finite element (FE) model was developed for predicting cutting forces and chip morphologies of polymers from the true stress–strain curve. A dual fracture process was used to simulate the cutting chip formation, incorporating both the shear damage failure criterion and the yield failure criterion, and considering the strain rate effect based on the Johnson–Cook formulation. The frictional behaviour between the cutting tool and specimen was defined by Coulomb’s law. Further, the estimated cutting forces and chip thicknesses at different nominal cutting depths were utilized to determine the fracture toughness of the polymer, using an existing mechanics method. It was found that the fracture toughness, cutting forces, and chip morphologies predicted by the FE model were consistent with the experimental results, which proved that the present FE model could effectively reflect the cutting process. In addition, a parametrical analysis was performed to investigate the effects of cutting depth, rake angle, and friction coefficient on the cutting force and chip formation, which found that, among these parameters, the friction coefficient had the greatest effect on cutting force.

## 1. Introduction

Orthogonal cutting is one of the most widely used machining methods for materials. During a cutting process, material is removed from a workpiece in the form of continuous or discontinuous chips, to obtain a designed geometry and surface finish. Predictions of the cutting force and chip morphology are important for improving cutting efficiency.

Extensive theoretical and experimental studies [1,2,3,4,5,6,7,8,9,10,11,12] have been presented to understand the process of metal cutting. For example, Merchant [1,2] presented a shear-angle model to describe the cutting process with a continuous chip. Lee et al. [3] developed a slip-line theory to explain the mechanism of the cutting process. Barry et al. [4] erformed an experiment to investigate the transition from continuous chip to saw-tooth chip formation. Over the last decades, numerical methods [13,14,15,16,17] have increasingly been used to investigate the complicated mechanisms involved with orthogonal metal-cutting, due to their high efficiency and low cost. For instance, Movahhedy et al. [13] developed an arbitrary Lagrangian–Eulerian FE model to simulate the metal cutting process. Wan et al. [14] presented an FE model to investigate the mechanism of dead metal zone formation during orthogonal cutting. Shi et al. [15] studied the effects of the friction coefficient on metal cutting. Zhang et al. [16] investigated the effects of cutting speed, feed rate, and rake angle on chip morphology transition using an FE analysis. Long et al. [17] performed an FE analysis to simulate burr formation of three types of metals.

The processes of polymer cutting are significantly different from the traditional cutting of metals in terms of chip formation, cutting forces, and heat transfer. This is because polymer material has a lower thermal conductivity compared with metals and many inorganic materials; therefore, heat build-up during machining can be a problem [18]. Furthermore, crack tip plasticity and creep behaviour of polymers during cutting conditions are completely different in metal materials. In recent years, orthogonal micro-cutting of polymers has drawn increasing attention [19,20,21,22,23,24,25]. Regarding the numerical method for orthogonal micro-cutting of polymers, Venu Gopala Rao et al. [19] developed an FE model to machine CFRP and GFRP composites. Wang et al. [20] presented a simple cutting method to predict the cutting force and determine fracture toughness of ductile polymers, whilst avoiding the problems associated with crack blunting. Mejías et al. [21] proposed a 2D FE model for orthogonal cutting of UD CFRP composite and accounted for the effect of various cutting parameters such as cutter rake, relief angle, cutting edge radius, and fibre orientation on machining-induced damage of composites. Yan et al. [22] established a link between machining parameter settings, burr formation, and cutting chip characteristics by performing dynamic mechanical analyses and fast micromilling experiments. A comprehensive review of these state-of-the-art machining responses suggests that though the damages, chip morphology, and cutting forces induced by cutting can be predicted accurately, the influence of the friction coefficient on the cutting force and chip formation is often very limited.

Although extensive experimental and numerical studies have been carried out for metal cutting, limited results are reported for micro-cutting of polymers using a numerical method. Moreover, few studies have focused on the prediction of cutting force during micro-cutting of polymers using true stress–strain curves. The aim of this paper is to develop a fully coupled FE model for simulating the cutting process, with a particular emphasis on the effects of the friction coefficient. A combination of shear damage failure and the yield failure criterion is used to simulate damage initiation, while material softening is described using a displacement-based approach. The strain rate effect is considered using the Johnson–Cook formulation. The tool-chip mechanical constraint formulation is regarded as kinematic contact method and is represented by Coulomb’s law. The effects of rake angle, cutting depth, and friction on the cutting force and chip morphologies are studied. The FE models are validated by comparing to the experimental cutting force data and optical measurements of chip morphologies.

## 2. Experiment

The orthogonal cutting experiments were performed on a CNC machine with a hydraulic power system (Minini Junior 90 M286, Used Machines, Pobegi, Slovenia) at room temperature; the schematic diagram is shown in Figure 1. A closed-loop feedback controller was embedded in the machine to achieve a positioning accuracy of ±100 nm. Two steel cutting tools were made from tungsten carbide with a clearance angle of 7° and a rake angle of 15° or 30°. The tip of the cutting tool was sharp with a radius of approximately 5 µm. The cutting tool was mounted on a three-axis load cell and the two cutting forces, *F_c_* and *F_t_*, were measured using a three-dimensional piezoelectric force transducer (Kistler 9257B, Winterthur, Switzerland) with a resolution of 0.02 N, which was attached to the hydraulic table (see Figure 1). The position of the tool was controlled using a calibrated screw thread. The specimen was supported between clamped steel plates on the servo-hydraulic testing machine. The workpiece material used in this article was high-density polyethylene (HDPE) machined into a prism shape with approximate dimensions of 10 × 10 × 7 mm^3^, and firmly clamped on the grinder using a holder. During orthogonal cutting, the cutting speed was set to 10 mm/min to minimize the strain rate effect and temperature effect. With such a low speed, all the cutting tests resulted in smooth, steady load traces, indicating that cutting reached steady-state equilibrium [20]. The nominal cutting depths were determined by CNC machine programming, and a more accurate measurement was performed using a chromatic distance sensor (Stil CHR 150), which had a measurement range of 0–300 μm, with a resolution of 80 nm. More cutting rig setup information can be found in literature [20].

Three common nominal cutting depths were chosen for this study: 60 μm, 120 μm, and 180 μm. Each cutting test was repeated at least five times to obtain an averaged cutting force. Before the cutting test, compression tests were performed using an Instron testing machine to measure the true stress–strain curves as shown in Figure 2. After the cutting test, chip formation profiles were detected using an optical microscope.

## 3. FE Modelling

In this section, a two-dimensional plane strain FE model was developed to predict the cutting force and chip formation of a polymer during orthogonal cutting, using commercial FE software ABAQUS. A schematic of the FE model for the orthogonal cutting process is shown in Figure 3. A cutting tool, with a rake angle of 15° or 30°, moved forward to machine the samples. Fixed constraints were applied on the bottom and two sides of the samples. The cutting tool was made of steel and had an elastic modulus of 210 GPa and a Poisson’s ratio of 0.3. The density was 7800 kg/m^3^. The samples were made of HDPE, and their true stress–strain curve used in the FE model is shown in Figure 2. The samples consisted of three parts, i.e., chip, failure layer, and the workpiece. The failure layer was designed for connecting the chips and workpiece and provided the energy to withstand the chip separation from the workpiece. A ductile damage criterion was used to simulate the chip formation.

As shown in Figure 4, the damage of the failure layer initiated at point B. When the equivalent plastic strain reached the fracture strain, the materials started to degrade, and damage evolution finished at point C. The energy during the damage evolution is *G*_c_. In this case, the fracture strain was set to 1.1 mm/mm [26], and *G*_c_ had a value of 1.25 kJ/m^2^ [27]. The interaction between chip and cutting tool is stick and slip. Frictions between the cutting tool and chip play a major role in the cutting process. Coulomb friction law was used between tool and interface, which yielded the relation:(1)$$\begin{array}{ll} \tau =\mu p & when\;\;\;\;\tau <\tau_{c}\\ \tau =\tau_{c} & when\;\;\;\;\tau \geq \tau_{c} \end{array}$$
where *μ* is the friction coefficient varying from 0.1 to 0.5 for parametric analysis, *p* is the normal pressure, and *τ_c_* is the critical friction stress. During the micro-cutting process, the cutting speed was relatively low, so the temperature effect on the properties of the materials can be neglected. A constant cutting speed with a value of 10 mm/s was used here to minimize the computational time. After sensitivity analysis, a total number of 7200 four-node plane strain elements (CPE4RT) were employed in this model, with a very fine mesh at the cutting path.

## 4. Discussion and Results

### 4.1. Validation of the FE Model

Orthogonal cutting of a polymer is a plane–strain problem [28]. Hence, specific cutting forces (i.e., *F_t_*/*b* and *F_c_*/*b*) are introduced in order to compare the cutting force of the specimen with different thicknesses. *F_t_* is the cutting force in the cutting direction, *F_c_* is the transverse force, and *b* the specimen thickness. Cutting tests were performed using cutting tools with two different rake angles as described in Section 2, and *F_t_*/*b* and *F_c_*/*b* were obtained as shown in Figure 5a,b, respectively. It was found that for both cutting tools, the *F_c_*/*b* increased as cutting depth increased, while *F_t_*/*b* decreased as cutting depth increased. The absolute values of *F_c_*/*b* and *F_t_*/*b*, however, both showed upward trends, which were directly related to the growth in volume of cutting materials [29]. In addition, the *F_c_*/*b* with a rake angle of 15° was greater than that with a 30° rake angle. Moreover, *F_t_*/*b* with a rake angle of 15° was positive when the cutting depth was low, and *F_t_*/*b* changed its direction when the cutting depth increased. In terms of cutting depth with a rake angle of 30°, the *F_t_*/*b* was negative even when the cutting force dropped to 60 μm.

After obtaining the *F_t_* and *F_c_*, the friction coefficient could be obtained using the following equation [30]:(2)$$\mu =\frac{F_{t}+F_{c}\tan\alpha }{F_{c}+F_{t}\tan\alpha }$$

Figure 6 shows that the friction coefficient decreased with an increase in cutting depth. Using the friction coefficient obtained in Figure 6, we predicted *F_t_*/*b* and *F_c_*/*b* using the FE model as shown in Figure 4. It was shown that the *F_c_*/*b* could be taken as a function of cutting depth by simulation and experiment with a rake angle of 15° and 30°. The *F_c_*/*b* and *F_t_*/*b* obtained by FE analysis were compared with those obtained experimentally at various cutting depths. We also examined the chip formation during cutting. Figure 7 shows that the chip curling determined by the FE model corresponds well with those obtained experimentally for all cases. When the cutting depth was low, the chip curling became obvious. Therefore, it is concluded that our FE model can be used to predict the cutting force of polymers.

### 4.2. Determination of Fracture Toughness Using Orthogonal Cutting

The experimental scheme required the measurement of cutting forces (*F_c_*/*b* and *F_t_*/*b*) for a cutting of width *b*. Wang et al. [20] proposed a method to estimate the fracture toughness and yielding stress of the polymer using a cutting method:(3)$$\frac{F_{c}}{b}-\frac{F_{t}}{b}\tan\phi =\frac{\sigma_{y}}{2}\left(\tan\phi +\frac{1}{\tan\phi }\right)h+G_{c}$$
(4)$$\phi =\frac{h\cos\alpha }{h_{c}-\cos\alpha }$$

If cutting forces (*F_c_* and *F_t_*) and chip thicknesses (*h_c_*) are measured in experiments over a range of cutting depth (*h*), *F_c_*/*b* − *F_t_*/*b*tan*ϕ* can be plotted against *h*(tan*ϕ* + 1/tan*ϕ*)/2; thus, the fracture energy, *G_c_*, required to create new surfaces can be obtained from the intercept. In such a process, the chip removal is equivalent to a fracture process where a crack is advancing at the tip of the cutting tool. The fracture energy, *G_c_*, should be the sum of both the surface energy for the new crack surfaces formed, and the plastic deformation energy around the advancing crack tip, based on concept of fracture mechanics. It is anticipated that the plastic zone size is dependent on the cutting depth, and in particular, the plastic zone size (≈*G_c_*/*σ_y_*) at the crack is confined by the cutting depth. When the cutting depth is reduced, the contribution of plastic deformation to energy, *G_c_*, reduces.

Figure 8 shows the fracture toughness obtained by orthogonal cutting at a speed of 10 mm/s. With different cutting depths, we could obtain the fracture toughness of the film, which were 1.173 kJ/m^2^ and 1.368 kJ/m^2^ using the cutting tool with a rake angle of 15° and 30°, respectively. These two values are comparable with the fracture toughness (1.25 kJ/m^2^) inputted in the FE model. The yielding stress derived from the analyses have the values of 46.89 and 33.85 MPa for the two cutting tools, which are higher than the yielding stress derived from compression test (27.4 MPa), although it is accepted that the strain rates in cutting and from the compression tests were not matched, which is consistent with the conclusion in the literature [31]. It can be found that workpiece materials have high strain rate sensitivity to cutting machining, even though the cutting speed was small, with a value of 10 mm/s. Therefore, it is recommended that the material strain rate effect (such as Johnson–Cook model) should be considered in the FE analysis for simulating cutting responses, even with a relatively small cutting speed.

### 4.3. Parametric Analysis

In this section, we performed a parametric analysis to investigate the effects of the friction coefficient and on the cutting force and chip formation. It has been demonstrated that thrust forces are intimately related to sub-surface damage, with observations that the less thrust force is achieved, the less induced damage is obtained [32]. Figure 9a shows *F_c_*/*b* with different friction coefficients and cutting depths, and shows that the *F_c_*/*b* increases with an increase in the friction coefficient. In addition, *F_c_*/*b* increased with increasing cutting depth. In terms of cutting force in the transverse direction, *F_t_*, *F_t_*/*b* increased with an increase in the friction coefficient as shown in Figure 9b. Interestingly, there is an intersection point for the three *F_t_*/*b* versus friction coefficient (*μ* ≈ 0.28) curves. The *F_t_*/*b* increased with an increase in cutting depth when *μ* ≤ 0.28, while this trend was the opposite when *μ* > 0.28. Furthermore, the value of *F_t_*/*b* was negative when the cutting depth = 180 μm and *μ* = 0.28. Regarding a rake angle of 30°, it was found that the *F_c_*/*b* was smaller than that with a low rake angle (Figure 8). Figure 10 shows that *F_c_*/*b* and *F_t_*/*b* increased with an increase in friction coefficient and cutting depth. The effect of friction coefficient on the *F_t_*/*b* was more significant when the value of cutting depth was small. *F_t_*/*b* was less than zero and the absolute value of *F_t_*/*b* decreased with an increase in friction coefficient. It also can be found from Figure 9 and Figure 10 that the slope of the *F_c_*/*b* versus friction coefficient curves increased with an increase in cutting depth, implying that the friction coefficient has a more significant effect on the *F_c_*/*b* for the larger cutting depth. The value of *F_t_*/*b* increased by 277% and 73% on average for the cutting with rake angles of 15° and 30°, respectively; while the value of *F_c_*/*b* increased by 58% and 45%, respectively. This implies that the friction coefficient has a greater impact on *F_t_*/*b* than *F_c_*/*b*. It was also found that the value of *F_c_*/*b* was relatively larger when induced by the cutting tool with a rake angle of 15°, which indicates this cutting tool can cause more subsurface damage compared to the tool with a rake angle of 30°.

Next, we examine the effects of friction coefficient on chip curling of HDPE with a rake angle of 15°, under different cutting depths. Figure 11, Figure 12 and Figure 13 show that chips tend to curl after cutting. The diameter of the chip curling increased with an increase in friction coefficient and cutting depth. The shear zone is evident in Figure 11, Figure 12 and Figure 13: a triangular shape (red) radiating from the tool tip to the free surface at the root of the chip. The Mises stresses at the triangular region have a relatively high value but have lower values in the vicinity of the tool edge because the material in this zone is softened by the significant temperature rise [14]. Furthermore, the chip formation and stress distributions of each HDPE cutting were almost the same. With increasing cutting depth and friction coefficient, the yield region will concentrate at the shear zone.

## 5. Conclusions

This study proposed a novel orthogonal cutting FE method for predicting the cutting force and chip morphologies of polymers. Shear damage failure and yield failure criterion from the true stress–strain curve were used to describe chip separation. The strain rate effect based on the Johnson–Cook formulation was included. The frictional behaviour of the tool-chip interface was defined by Coulomb’s law. In addition, values of fracture toughness and yield strength were determined for HDPE using standard tests for comparison with the values obtained from cutting. The experimental work was performed to verify the FE model. The following conclusions can be drawn:The value of *F_c_*/*b* increases with the increase in cutting depth, while *F_t_*/*b* show an opposite trend. The cutting forces induced by the cutting tool with a rake angle of 30° are smaller than those with a rake angle of 15°, which means improved surface cutting quality can be produced by a cutting tool with a rake angle of 30°.The friction coefficient has more significant effects on cutting force (*F_t_*/*b* and *F_c_*/*b*) than that of cutting depth and rake angle. The value of *F_t_*/*b* increased by 277% and 73% on average for the cutting with rake angles of 15° and 30°, respectively, and the value of *F_c_*/*b* increased by 58% and 45%, respectively. This implies that the friction coefficient has a greater impact on *F_t_*/*b* than *F_c_*/*b*. Hence, the machining quality of the composite materials can be greatly improved by reducing the friction force.The chip formations and stress distributions of each HDPE cutting are almost the same. The diameter of the chip curling increase with an increase in friction coefficient and cutting depth. The stresses at the triangular region have a relatively high value but have lower values in the vicinity of the tool edge.The yielding stress during cutting, at a speed of 10 mm/s, is a factor that is 1.7 times higher than the quasi-static compression test value. Hence, it is recommended that the material strain rate effect should be considered in the orthogonal cutting FE model, even when the cutting speed is relatively small.

Orthogonal cutting often causes severe subsurface damages to composite materials around the cutting face. Cutting-induced damage analysis can provide guidance and information regarding the optimization of cutter parameters and cutting technology for reducing damage to workpiece surfaces. In future studies, an FE model that can account for various damage patterns will be developed and validated using experimental cutting force data as well as non-destructive testing of machining-induced damages.

## Figures and Tables

**Figure 1 polymers-14-00189-f001:**
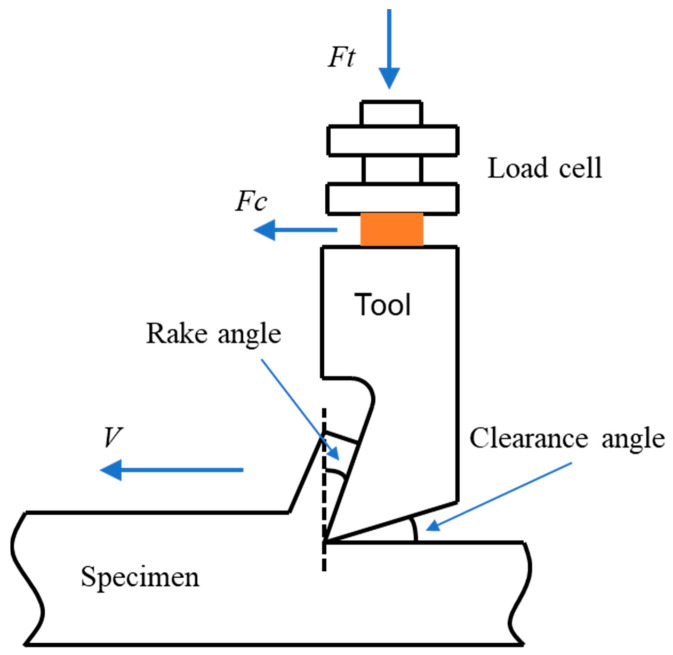
Schematic diagram of the orthogonal cutting rig setup.

**Figure 2 polymers-14-00189-f002:**
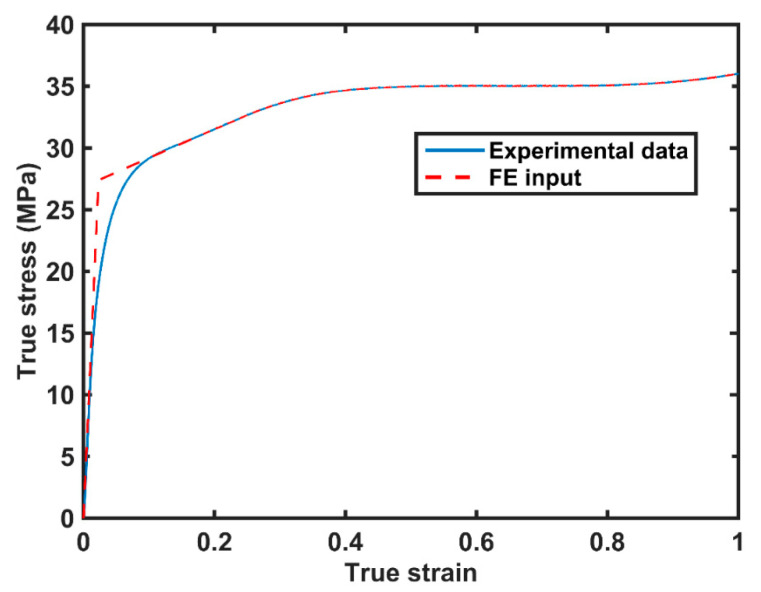
True stress–strain curves obtained by experiment and FE analysis.

**Figure 3 polymers-14-00189-f003:**
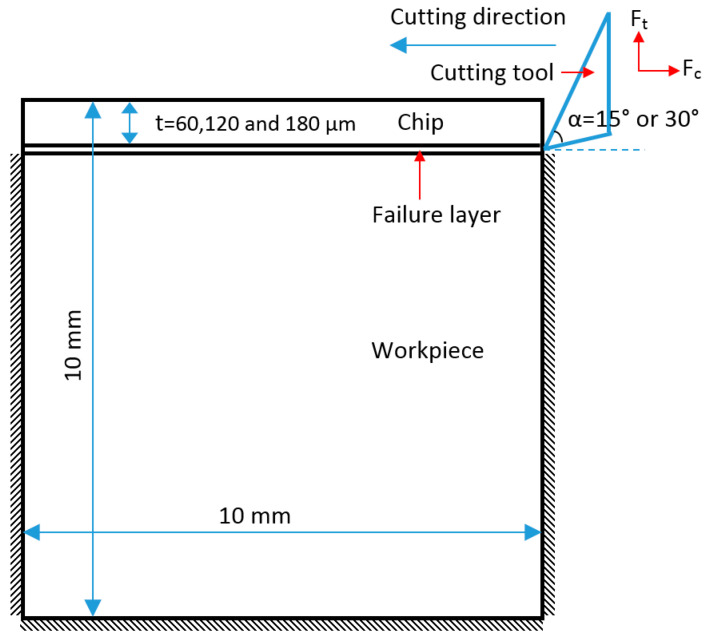
Schematic of geometry and boundary conditions in FE analysis.

**Figure 4 polymers-14-00189-f004:**
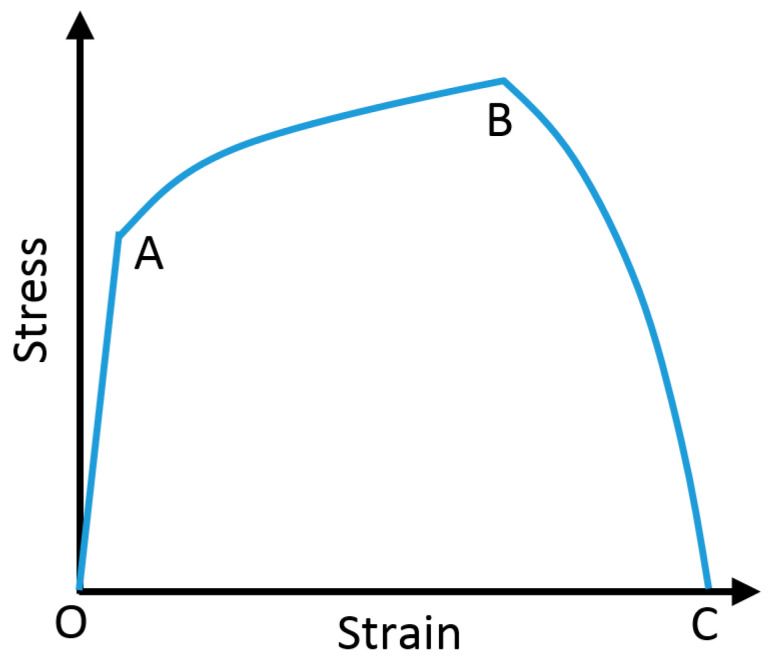
Schematic of damage initiation and evolution in HDPE.

**Figure 5 polymers-14-00189-f005:**
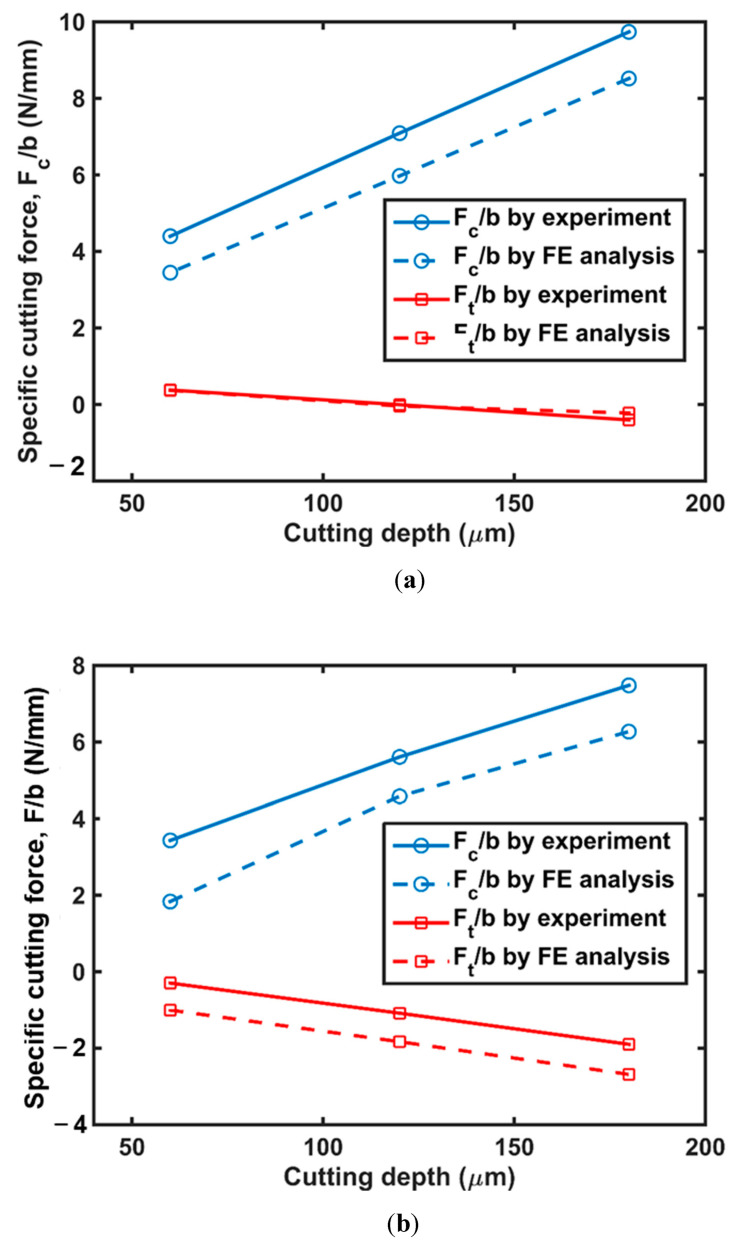
Comparison of cutting force obtained by experiment and simulation of HDPE with a rake angle of (**a**) 15° and (**b**) 30°.

**Figure 6 polymers-14-00189-f006:**
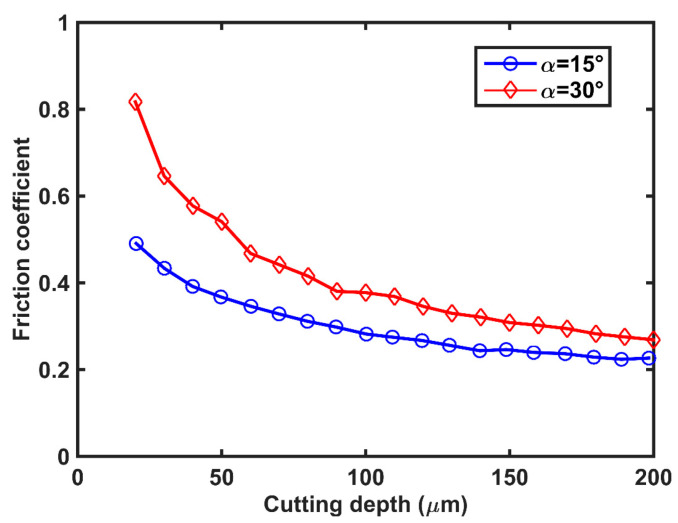
Friction coefficient as a function of cutting depth in HDPE.

**Figure 7 polymers-14-00189-f007:**
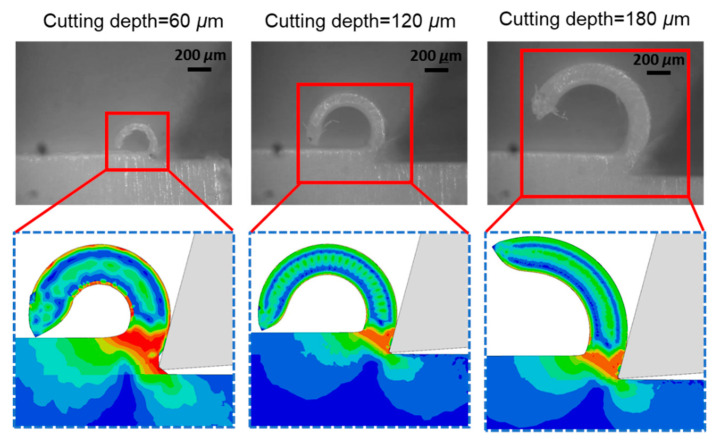
Chip formation of the polymer under orthogonal cutting with a 15° cutting tool.

**Figure 8 polymers-14-00189-f008:**
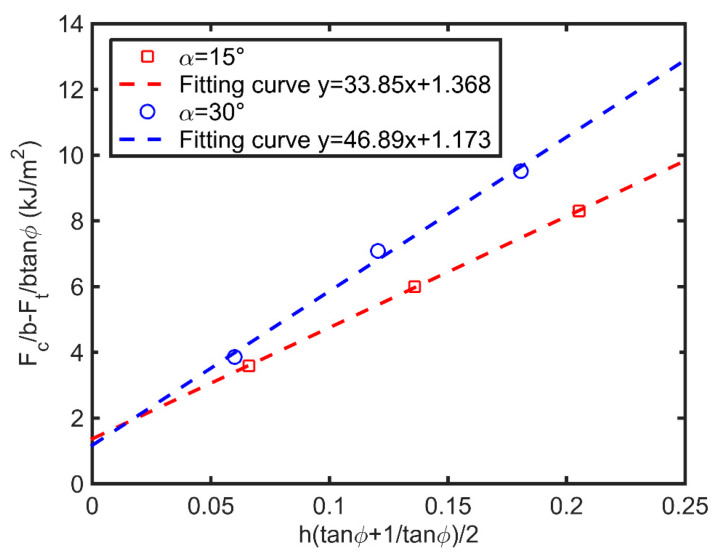
Fracture toughness obtained by orthogonal cutting.

**Figure 9 polymers-14-00189-f009:**
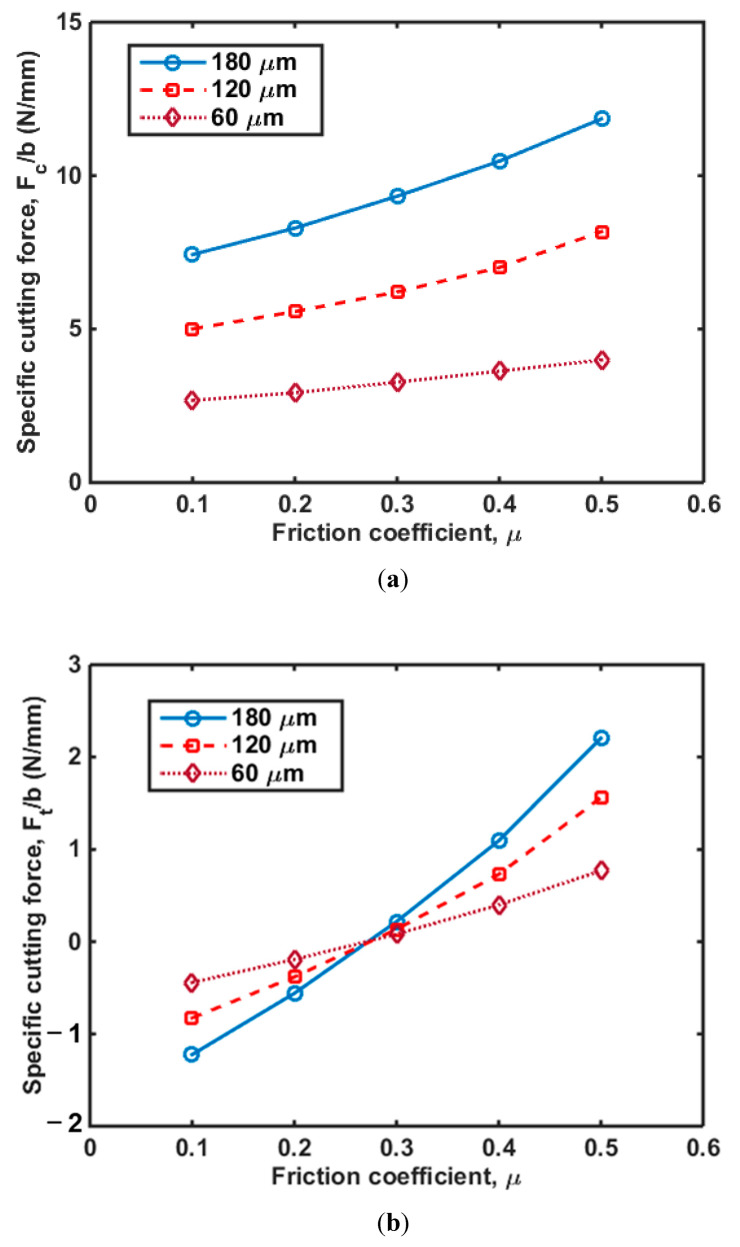
Cutting force in (**a**) cutting direction and (**b**) transverse direction as a function of friction coefficient in HDPE with a rake angle of 15°.

**Figure 10 polymers-14-00189-f010:**
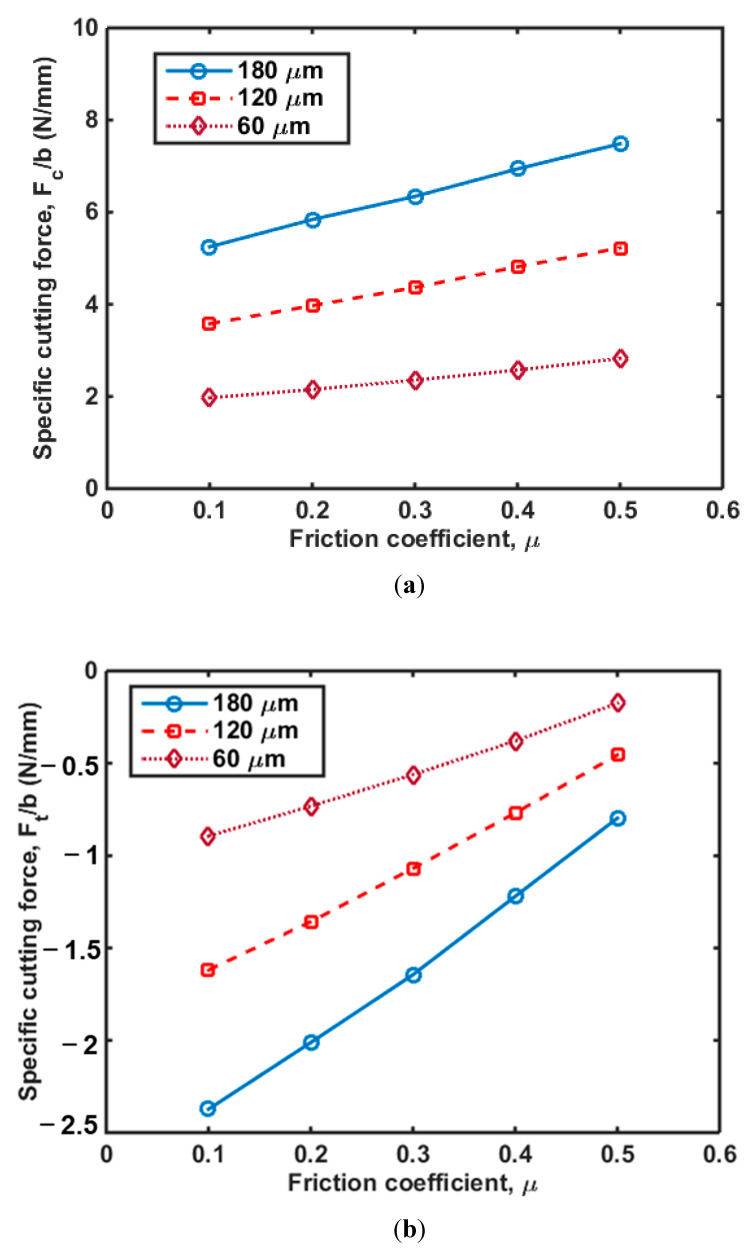
Cutting force in (**a**) cutting direction and (**b**) transverse direction as a function of friction coefficient in HDPE with a rake angle of 30°.

**Figure 11 polymers-14-00189-f011:**
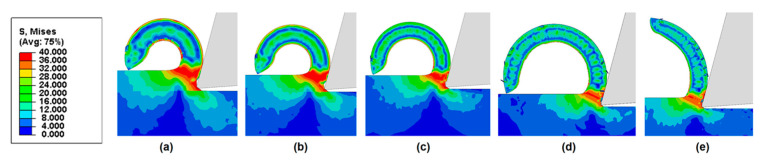
Chip morphologies (cutting depth = 60 μm) predicted by FE model with a rake angle of 15° when friction coefficient is (**a**) 0.1, (**b**) 0.2, (**c**) 0.3, (**d**) 0.4, and (**e**) 0.5.

**Figure 12 polymers-14-00189-f012:**
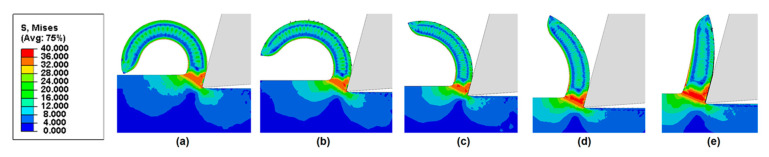
Chip morphologies (cutting depth = 120 μm) predicted by FE model with a rake angle of 15° when friction coefficient is (**a**) 0.1, (**b**) 0.2, (**c**) 0.3, (**d**) 0.4, and (**e**) 0.5.

**Figure 13 polymers-14-00189-f013:**
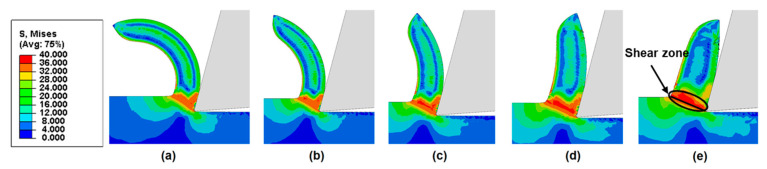
Chip morphologies (cutting depth = 180 μm) predicted by FE model with a rake angle of 15° when friction coefficient is (**a**) 0.1, (**b**) 0.2, (**c**) 0.3, (**d**) 0.4, and (**e**) 0.5.

## Data Availability

Not applicable.

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
