# Peer review of "Prediction of Cutting Force and Chip Formation from the True Stress–Strain Relation Using an Explicit FEM for Polymer Machining"

_polymers, 2022, doi:10.3390/polym14010189_

Round 1

Reviewer 1 Report

The paper follows the methods used for modeling of cutting in metallic materials and the novelty is rather limited. As such, the results are also trivial. 

The authors need to improve their manuscript in terms of originality, explain the selection of parameters and the use of the specific modeling methods that are rather old and outdated.

Author Response

Thank you for the suggestion. We have made a lot of changes to the article, and research significance, innovation and the important conclusions have been described in detail. Please check the modifications in the revised manuscript which are highlighted in red colour.

Reviewer 2 Report

Reviewed article concerns prediction of cutting force and chip formation from the true stress-strain relation using an explicit FEM for polymer machining and is write in accordance with generally accepted standards of the scientific works. After careful reading of the submitted text there are some substantive remarks that should be taken into consideration by the Authors to improve reviewed text.

  1. The abstract should include information about new methods, results, concepts, and conclusions – in its current form, the abstract needs to be rewritten to include more precise information on achievements described in the manuscript.
  2. Literature review should be improved providing more references to recent works from the area of described study.
  3. At the end of the introduction should be clearly and concise given the research gap to create the appropriate lead up for the motivation of the work and the novelty of given approach should be also emphasized.
  4. The Authors should more carefully and more detailed describe methodology of experiments.
  5. I suggest providing more precise information about used experimental and measurement positions.
  6. Presented study widely covers defined scientific problem and with experimental investigations provides proper background for given conclusions, however deeper scientific consideration of obtained results referred to the basic phenomena in cutting processes should be given.
  7. In the discussion section should be provide more references to already known results from literature.
  8. The strengths and limitations of the obtained results and applied methods should be clearly described.
  9. The main conclusions should refer to specific values (results of analysis) as well as basic phenomena that cause described results.
  10. The conclusion should be improved in term of the new knowledge gained during analysis, which should be concise with the journal scope.

Round 2

Reviewer 1 Report

Although the authors have provided some modifications, the core of the analysis remains the same and the results that are presented are not novel nor present any practical interest. Coulomb's law for the evaluation of friction and two dimensional models for forces require much more justification. The fact that the analysis is carried out on polymers is not enough novelty in my opinion.

Author Response

Thank you for the suggestion. Even if the method used in this study is similar to metal cutting research, there are still many difference and novelty. Compared to metal materials and many inorganic materials, the polymer material has a low thermal conductivity, then heat build-up during machining would be a tough problem. Furthermore, crack tip plasticity and creep behaviour of polymers during cutting conditions is totally different from metal materials. In this study, we developed an explicit finite element (FE) model for predicting cutting forces and chip morphologies of polymers from the true stress-strain curve. The cutting theory is employed to calculate the amount of fracture energy dissipated in the cutting process. In addition, the estimated cutting forces and chip thicknesses at different nominal cutting depths were utilized to determine the fracture toughness of the polymer using an existing mechanics method. Finally, a parametrical analysis was performed to investigate the effects of cutting depth, rake angle, and friction coefficient on the cutting force and chip formation, which finding that the friction coefficient has the greatest effects on cutting force among these parameters.

Reviewer 2 Report

All my comments were taken into consideration. The article can be accepted for publication in it's present form.

Author Response

Thank you very much for your recognition of our work

Round 3

Reviewer 1 Report

The paper can be published in its present form.